# Ca_V2.1 channel mutations causing familial hemiplegic migraine type 1 increase the susceptibility for cortical spreading depolarizations and seizures and worsen outcome after experimental traumatic brain injury

**Nicole A Terpolilli**[1,2,3], **Reinhard Dolp**[2], **Kai Waehner**[4], **Susanne M Schwarzmaier**[1,3,5], **Elisabeth Rumbler**[2], **Boyan Todorov**[6], **Michel D Ferrari**[7], **Arn MJM van den Maagdenberg**[6,7], **Nikolaus Plesnila**[1,3]*

[1]Institute for Stroke and Dementia Research, Munich University Hospital, Munich, Germany; [2]Department of Neurosurgery, Munich University Hospital, Munich, Germany; [3]Munich Cluster for Systems Neurology (SyNergy), Munich, Germany; [4]Department of Neurosurgery, Mannheim University, Mannheim, Germany; [5]Department of Anesthesiology, Munich University Hospital, Mannheim, Germany; [6]Department of Human Genetics, Leiden University Medical Center, Leiden, Netherlands; [7]Department of Neurology, Leiden University Medical Center, Leiden, Netherlands

*For correspondence:
nikolaus.plesnila@med.uni-muenchen.de

**Competing interest:** The authors declare that no competing interests exist.

**Abstract** Patients suffering from familial hemiplegic migraine type 1 (FHM1) may have a disproportionally severe outcome after head trauma, but the underlying mechanisms are unclear. Hence, we subjected knock-in mice carrying the severer S218L or milder R192Q FHM1 gain-of-function missense mutation in the *CACNA1A* gene that encodes the $\alpha_{1A}$ subunit of neuronal voltage-gated Ca_V2.1 (P/Q-type) calcium channels and their wild-type (WT) littermates to experimental traumatic brain injury (TBI) by controlled cortical impact and investigated cortical spreading depolarizations (CSDs), lesion volume, brain edema formation, and functional outcome. After TBI, all mutant mice displayed considerably more CSDs and seizures than WT mice, while S218L mutant mice had a substantially higher mortality. Brain edema formation and the resulting increase in intracranial pressure were more pronounced in mutant mice, while only S218L mutant mice had larger lesion volumes and worse functional outcome. Here, we show that gain of Ca_V2.1 channel function worsens histopathological and functional outcome after TBI in mice. This phenotype was associated with a higher number of CSDs, increased seizure activity, and more pronounced brain edema formation. Hence, our results suggest increased susceptibility for CSDs and seizures as potential mechanisms for bad outcome after TBI in FHM1 mutation carriers.

## Editor's evaluation

This paper will be of considerable interest to familial hemiplegic migraine type 1 (FHM) sufferers who may experience traumatic brain injury (and their physicians), as well researchers with an interest in the spectrum and phenotypic consequences of mutations in the voltage-gated, P/Q-type Ca²⁺ channel, *CACNA1A*. The authors demonstrate that patients carrying a gain-of-function S218L

missense mutation in *CACNA1A* exhibit a gene-dosage-dependent increase in the susceptibility to cortical spreading depolarization (CSD), seizure activity, and brain edema formation following TBI.

## Introduction

Familial hemiplegic migraine (FHM) is a rare monogenic subtype of migraine with aura caused by mutations in various genes (*Ferrari et al., 2015*). FHM type 1 (FHM1) is caused by specific missense mutations in *CACNA1A* that encodes the $\alpha_{1A}$ subunit of neuronal voltage-gated $Ca_V2.1$ (P/Q-type) $Ca^{2+}$ channels (*Ophoff et al., 1996*; *Ferrari et al., 2015*). While patients carrying the R192Q mutation display only hemiplegic migraine, those with the S218L mutation show additional clinical features, such as cerebellar ataxia and seizures (*Ferrari et al., 2015*). In patients with the S218L mutation, even trivial head trauma was shown to trigger severe, sometimes fatal attacks that are associated with seizures, coma, and massive brain edema (*Fitzsimons and Wolfenden, 1985*; *Ophoff et al., 1996*; *Kors et al., 2001*; *Stam et al., 2009*; *Ferrari et al., 2015*). The mechanisms underlying this phenotype have, however, not been identified.

Experiments performed in transgenic knock-in mice expressing the human FHM1 S218L $Ca_V2.1$ channel mutation (*van den Maagdenberg et al., 2010*) exhibit increased neuronal $Ca^{2+}$ influx, enhanced cortical glutamatergic synaptic transmission, and an increased susceptibility to experimentally induced cortical spreading depolarizations (CSDs) (*Eikermann-Haerter et al., 2009*; *van den Maagdenberg et al., 2010*; *Chanda et al., 2013*; *Ferrari et al., 2015*; *Vecchia et al., 2015*). Since CSDs have been implicated in swelling of neuronal cells in healthy brain and were suggested to be involved in brain edema formation under pathological conditions (*Takano et al., 2007*; *Seidel et al., 2016*; *Hartings et al., 2017*), we hypothesized that a higher susceptibility to CSDs may result in overshooting edema formation and subsequent worsen outcome after traumatic brain injury (TBI) of FHM1 mutation carriers. To investigate this hypothesis, we exposed the two FHM1 mouse models carrying either the severer S218L or the milder R192Q $Ca_V2.1$ channel mutation and respective controls to experimental TBI and recorded CSDs together with brain edema formation, intracranial pressure (ICP), lesion volume, and neurological function.

## Results

### Induction of CSDs in wild-type mice

To demonstrate the functionality of our CSD recording setup using epicranial Calomel electrodes, we first induced CSDs in healthy wild-type (WT) mice by topical application of 2 μl 0.1 M KCl onto the cortex while simultaneously recording CSD events. CSDs were identified by a reduced amplitude of the electroencephalogram (EEG) signal and a negative shift in the direct current (DC) potential followed by a compensatory increase in local cerebral blood flow (CBF) (*Figure 1C*). Following TBI, we observed similar changes, that is a depression of the EEG and a negative shift of the DC signal, while CBF increased over both cerebral hemispheres (*Figure 1D*) indicating that brain trauma, unlike KCl application, induces bilateral CSDs, as previously described (*von Baumgarten et al., 2008*). Thus, using the currently used experimental setup we were able to record KCl induced as well as post-trauma CSDs.

### CSD recordings before and after TBI in FHM1 mutant mice

After finishing the surgical preparation (skin incision, removal of the periosteum, drilling) and placing all measurement probes, baseline recordings for DC potential, CBF, and EEG were performed for 15 min. No CSDs were recorded during this period of time. Thereafter, the bone flap was removed in order to apply the trauma to the cerebral cortex. After removal of the bone flap, which is associated with (minor) mechanical stress to the dura mater, one or several CSDs were recorded in the ipsilateral hemisphere depending of the genotype of the animals. While only one CSD was observed in 2/18 investigated WT mice ($n$ = 8 S218L WT littermates and $n$ = 10 R192Q WT littermates) and in 3/10 heterozygous S218L mutant mice, all homozygous FHM1 mutant mice ($n$ = 7 S218L and $n$ = 8 R192Q mutant mice) showed at least one CSD. In homozygous S218L mutant mice even trains of up to four CSDs were observed after removal of the bone flap. In total 2, 3, 8, and 16 CSDs (0.1, 0.4, 1.3, and 2.3 CSDs/animal) were observed in WT, heterozygous S218L, homozygous R192Q, and homozygous

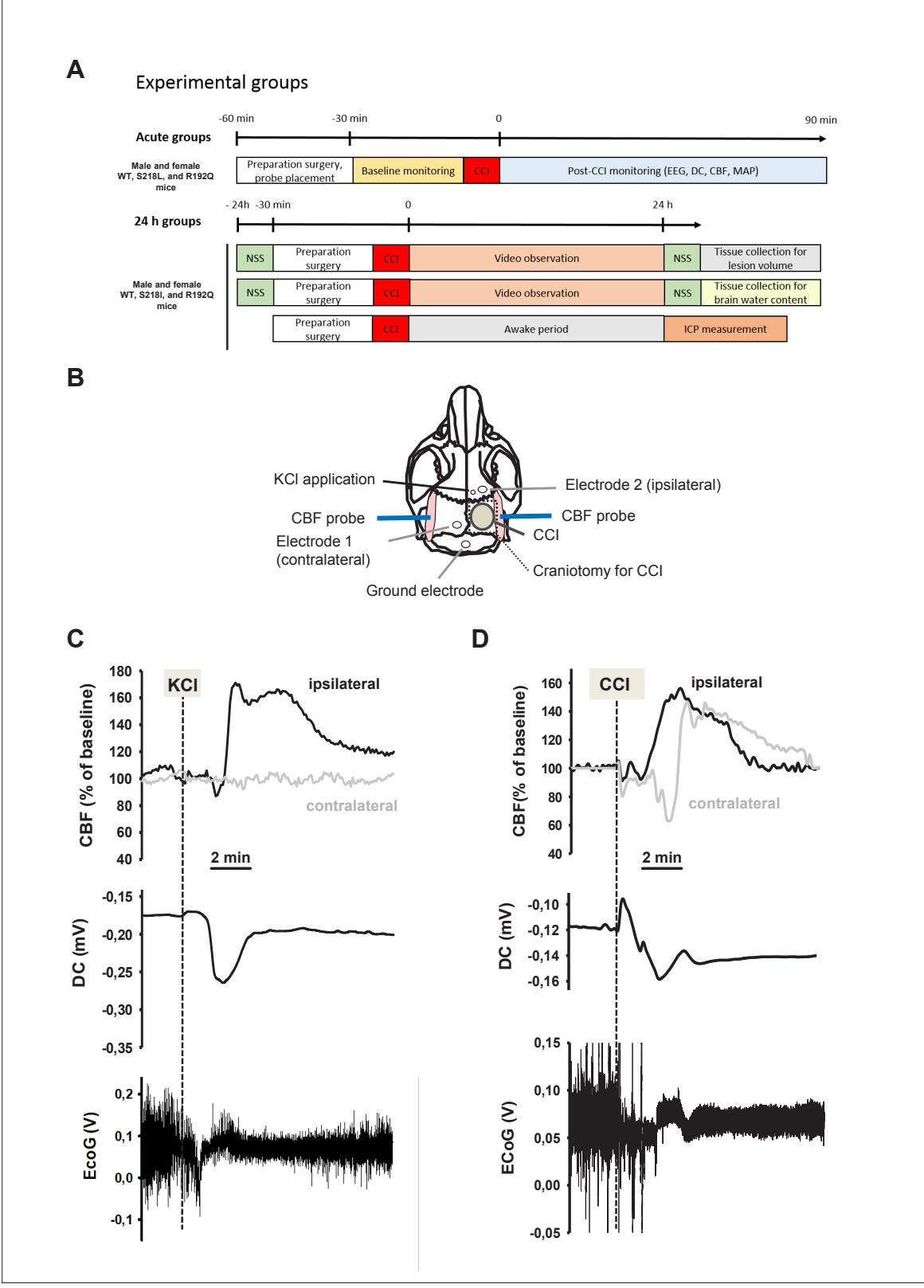

**Figure 1.** Design and methodology. (**A**) Experimental plan. (**B**) Electrode placement. (**C**) Multiparametric recording of a cortical spreading depolarization (CSD) event elicited by application of 0.1 M potassium chloride (KCl): CSD is characterized by a negative direct current (DC) shift (middle panel, arrow), narrowing of electroencephalogram (EEG) amplitude (lower panel), and a consecutive cerebral hyperperfusion as indicated by an increase of cerebral blood flow (CBF) measured over the right hemisphere (ipsilateral to KCL application); CSD is restricted to the ipsilateral side, CBF on the contralateral

*Figure 1 continued on next page*

*Figure 1 continued*

hemisphere is not altered during CSD. (**D**) After controlled cortical impact (CCI), the DC shift is bidirectional, resulting in a pronounced change of EEG amplitude and is followed by a marked transient increase of cerebral perfusion first in the ipsilateral, then in the contralateral hemisphere.

S218L mutant mice, respectively. No further CSD activity was observed thereafter. These data indicate, that mechanical stress to the dura mater alone induced CSDs in 10% of WT mice, but in 100% of homozygous FHM1 mutant mice and that heterozygous and homozygous FHM1 mutant mice had four to eight times more CSDs than WT littermate controls, respectively. This finding suggests that FHM1 mutant mice have a higher, gene-dose-dependent (het S218L < hom. R192Q < hom. S218L) susceptibility to CSDs induced by mechanical stress to the dura mater as compared to their WT littermates.

After another 15 min of baseline recordings without further CSDs TBI was induced. All investigated animals generated one bihemispheric CSD immediately after the impact independent of their genotype as evidenced by a negative shift in the DC potential, a depression of the amplitude of the EEG signal, and a sequential increase of CBF in both hemispheres (*Figure 2A*) as already shown for WT mice (*Figure 1D*) and as previously published (*von Baumgarten et al., 2008*). After this initial bihemispheric CSD, only five additional CSDs were observed in the traumatized hemisphere of WT mice (*n* = 18; 0.2 CSDs/mouse/hr), findings well in line with previously published data from our laboratory (*von Baumgarten et al., 2008*). In contrast, all FHM1 mutant mice generated numerous post-trauma CSDs. In total 22, 37, and 79 CSDs (1.5, 2.5, and 7.5 CSDs/animal/hr) were observed in homozygous R192Q, heterozygous S218L, and homozygous S218L mutant mice, respectively (*Figure 2B*). Hence, homozygous R192Q, heterozygous S218L, and homozygous S218L mice generated 4, 7, and 16 times more CSDs after TBI than WT animals, respectively (p < 0.001 S218L hom. vs. S218L WT and p < 0.01 R192Q hom. vs. R192Q WT). Further, FHM1 mutant mice generated additional bihemispheric CSDs after TBI. While only two bilateral CSDs were observed in WT mice later than 5 min after TBI, 3, 9, and 29 bilateral CSDs were recorded in homozygous R192Q, heterozygous S218L, and homozygous S218L mutant mice, respectively. Heterozygous and homozygous S218L mutant mice also generated CSDs in the contralateral hemisphere (6 and 9, respectively), types of CSDs which we never observed in WT or R192Q mutant mice, the strain which carries the less severe FHM1 Ca$_V$2.1 channel mutation. Consequently, FHM1 mutant mice had a gene-dose-dependent several fold higher susceptibility for post-TBI CSDs than their WT littermates.

## Posttraumatic seizure activity

During the CSD and EEG recordings after TBI we observed intermittent and synchronized EEG activity with a frequency of 4–8 Hz in homozygous S218L mutant mice (*Figure 3A*) in parallel with tonic contraction of the tail and/or opisthotonus indicative of seizure activity. To characterize and quantify posttraumatic seizure activity systematically, a second cohort of heterozygous and homozygous S218L mutant mice and their WT littermates were followed by continuous video monitoring for 24 hr after TBI (*n* = 16, 17, and 18, respectively). Seizures were identified by either tonic or clonic contractions of the upper or lower limb contralateral to the traumatized hemisphere or by generalized tonic or clonic contractions. While only one seizure was observed in a single WT animal, all heterozygous and homozygous S218L mutant mice had at least one seizure. Some seizures (1.0 ± 1.3) were observed in heterozygous and 2.5 ± 5.7 seizures in homozygous S218L mutant mice (*Figure 3B*). In an additional cohort of WT, heterozygous, and homozygous S218L mice (*n* = 11 per group) we assessed the duration of seizures. While seizures in affected WT mice lasted only a few seconds (0.0 ± 0.4 min; *Figure 3C*), the cumulative seizure duration per animal was longer in heterozygous (2.0 ± 2.5 min), and much longer in homozygous S218L mutant mice (72 ± 44 min). While none of the 29 investigated WT mice and only one out of 28 investigated heterozygous S218L mutant mice died during seizures (3.6%), 21 out of 49 investigated homozygous S218L mutant mice (43%) died during a generalized tonic–clonic seizure after TBI.

## Brain edema and ICP after TBI

Unhandled WT and FHM1 mice had a normal brain water content of around 77% before TBI (*Figure 4A, B* – naive). Sham-operated animals had no significant brain edema formation, while in traumatized mice brain water content increased by 3–4% in the traumatized and by 1.5–2% in the contralateral hemisphere 24 hr after TBI (*Figure 4*, 4 hr after trauma, light bars), values comparable to previously

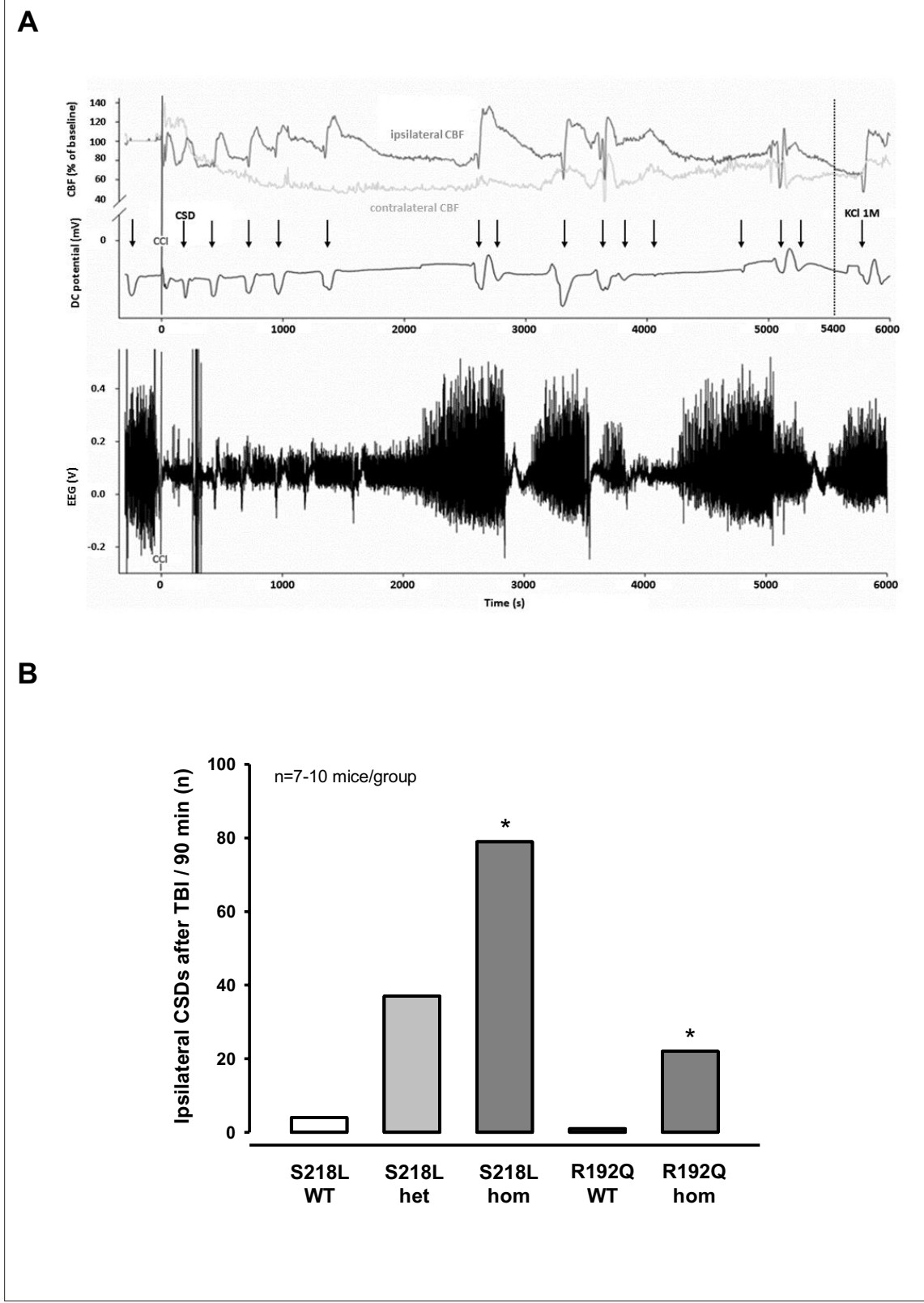

**Figure 2.** Cortical spreading depolarizations (CSDs) and electroencephalogram (EEG) in familial hemiplegic migraine type 1 (FHM1) mutant mice after trauma. (**A**) Exemplary multiparametric recording of a (male) homozygous S218L mutant mouse showing multiple posttraumatic CSDs (black arrows). After the end of the observation period (90 min post controlled cortical impact (CCI), 5400 s), 2 µl 0.1 M KCl was applied to trigger a CSD event. The majority of posttraumatic CSDs were recorded ipsilateral to CCI as indicated by a hyperperfusion over the ipsilateral cortex (dark gray cerebral blood

*Figure 2 continued on next page*

*Figure 2 continued*

flow [CBF] trace). However, in some instances, sequential hyperperfusion of both hemispheres (corresponding to a bihemispheric CSD) or an isolated contralateral hyperperfusion (corresponding to CSD originating in the contralateral, i.e., nontraumatized side) was observed after direct current (DC) and EEG changes characteristic for a CSD event. (**B**) After CCI, all mutant mice show a highly increased propensity for CSDs: S218L mutants showed a much higher frequency of CSD events during the 90 min observation time as compared to wild-type (WT) littermates (p < 0.001 S218L hom. vs. S218L WT, analysis of variance [ANOVA] on ranks). Also in the R192Q strain, homozygous mice experienced more CSDs than WT mice (p < 0.01 R192Q hom. vs. R192Q WT, ANOVA on ranks). S218L WT: n = 8; S218L het.: n = 10; S218L hom.: n = 7; R192Q WT: n = 10; R192Q hom.: n = 10.

The online version of this article includes the following source data for figure 2:

**Source data 1.** *Figure 2B*- Ipsilateral CSDs after TBI.

published data from our laboratory (*Zweckberger et al., 2006*; *Trabold et al., 2010*). Homozygotes of both FHM1 mutant mouse lines developed significantly more severe brain edema than WT controls (*Figure 4A, B*, 24 hr after trauma, dark bars). Brain edema formation led to an increase in ICP in all investigated groups (*Figure 4C*). Homozygotes of both FHM1 mutant lines had a significantly more pronounced intracranial hypertension than WT mice (*Figure 4C*).

Since an increase in ICP reduces cerebral perfusion pressure thereby aggravating pericontusional ischemia, brain edema formation may also result in additional tissue injury after TBI. Indeed, homozygous S218L mutant mice showed a significantly higher lesion volume (23.0 ± 3.7 mm$^3$) as compared to WT controls 24 hr after TBI (19.0 ± 1.9 mm$^3$; p < 0.02; *Figure 5A*). Larger lesion volumes translated into worse neurological function in homozygous S218L mutant mice (*Figure 5B*). In line with these findings, homozygous S218L mutant mice also showed a pronounced weigh loss as a sign of a reduced physical condition (*Figure 5C*).

No major differences were observed between male and female animals (data not shown).

## Discussion

The aim of the study was to identify possible mechanisms responsible for worse outcome after brain injury as this can sometimes be a severe consequence in patients suffering from FHM1. That is, patients with the S218L missense mutation suffer from hemiplegic migraine including cerebellar ataxia, epilepsy, and sometimes fatal attacks upon (mild) head trauma, while many patients with other *CACNA1A* mutations, like the R192Q mutation, suffer from hemiplegic migraine without an apparent risk for such additional clinical features (*Ferrari et al., 2015*). Therefore, we subjected knock-in mice that expressed either of the two FHM1 gain-of-function missense mutations in the α$_{1A}$ subunit of neuronal Ca$_V$2.1 Ca$^{2+}$ channels to TBI and investigated CSDs, the electrophysiological correlate of the migraine aura. Homozygous mutant mice with the more severe S218L mutation showed 20 times more CSDs upon mechanical stimulation of the dura mater before TBI and almost 20 times more CSDs after TBI. Furthermore, homozygous S218L mutant mice had long-lasting generalized seizures after TBI, a high, seizure-associated posttraumatic mortality of 60%, more brain edema formation, higher ICP values, larger lesion volumes, and worse functional outcome after TBI as compared to heterozygous mice from the same mouse line, homozygous mice expressing the milder R192Q mutation, or WT mice. Hence, the current results suggest that the increased susceptibility of FHM1 mutant mice to mechanically induced CSDs may explain the worse outcome of human S218L mutation carriers after TBI.

FHM1 mutant mice exhibit increased susceptibility to CSD induced by electrical stimulation (*van den Maagdenberg et al., 2004*; *van den Maagdenberg et al., 2010*) or topical application of KCl to the cerebral cortex (*Eikermann-Haerter et al., 2009*). So far, it was unclear whether FHM1 mutant mice also have more CSDs after a traumatic insult (and mechanical stimulation because of the surgery) to the brain. In the present study, we observed CSDs in homozygous S218L and R192Q mutant mice in association with removing the bone flap when performing a craniotomy. Removing of the bone flap is always associated with some mechanical stress to the dura mater and to minimal, short-term bleeding from ruptured bridging veins. While this minimal invasive procedure is well tolerated by WT mice, it induced a significant number of CSDs right after the removal of the bone flap in FHM1 mutant mice. Of note, no spontaneous CSDs were observed in either genotype neither before nor later than 5 min after removal of the bone flat. This demonstrated an increased susceptibility of FHM1 mutant mice to CSDs due to mild mechanical stress to the dura mater. Since CSD is the electrophysiological correlate

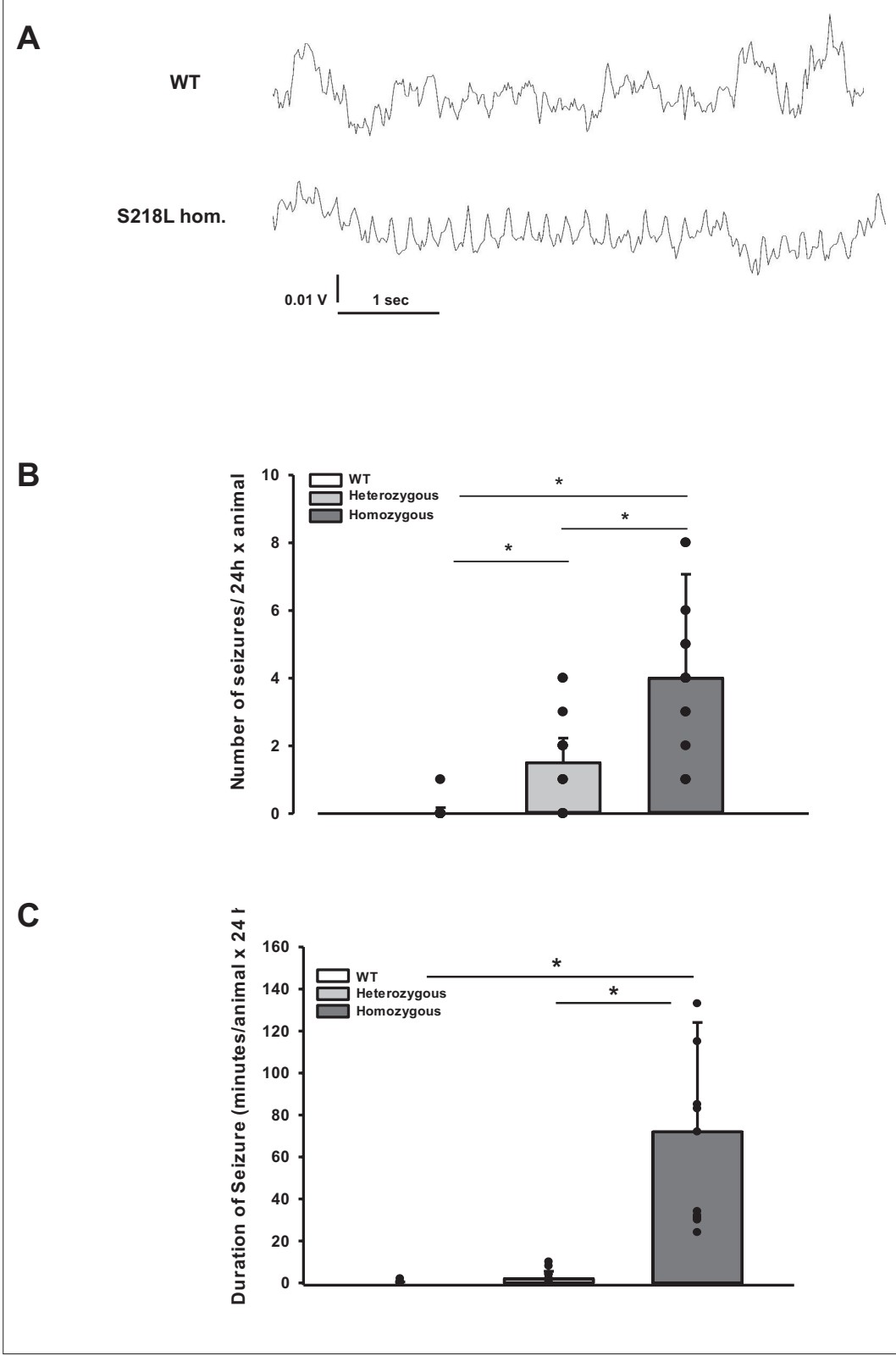

**Figure 3.** Posttraumatic seizures in familial hemiplegic migraine type 1 (FHM1) mutant mice after controlled cortical impact (CCI). (**A**) Exemplary sections of the electroencephalogram (EEG) trace of a homozygous S218L mutant mouse: While EEG showed a normal pattern during baseline monitoring (upper curve), synchronized EEG activity with a frequency of 4–8 Hz as shown in the lower trace was recorded in S218L mutants only; during these

*Figure 3 continued on next page*

*Figure 3 continued*

phases mice intermittently showed tonic tail contraction and/or opisthotonus suggestive for seizure activity. (**B**) While seizures occurred very rarely in wild-type (WT) mice (one event in total group), S218L mutants experienced a high number of seizures. The total number of epileptic seizures per 24 hr was significantly higher in homozygous S218L mutant than in heterozygous S218L mutant and WT mice ($n$ = 16–18 each; median ± 95% confidence interval, *p < 0.001, analysis of variance [ANOVA] on ranks). (**C**) Average duration of seizure activity in minutes per animal within the first 24 hr after CCI. Homozygous S218L mutants on average experienced 72 min of seizure activity in 24 hr while the few seizures occurring in heterozygous mutant and WT mice lasted significantly shorter ($n$ = 11 each; median ± 95% confidence interval, *p ≤ 0.001, ANOVA on ranks).

The online version of this article includes the following source data for figure 3:

**Source data 1.** *Figure 3B, C* Number and duraration of seizures after TBI.

of the migraine aura, these findings corroborate the phenotype of FHM1 in humans, that is attacks of migraine with aura after mechanical stress to the brain.

After TBI FHM1 mutant mice showed an up to 20 times higher rate of CSDs in the traumatized hemisphere, a higher seizure activity, more brain edema formation, and a worse outcome as compared to WT littermates. CSDs have previously been detected in TBI patients (**Mayevsky et al., 1996**; **Strong et al., 2002**; **Fabricius et al., 2006**) and in animal TBI models (**Takahashi et al., 1981**; **Ozawa et al., 1991**; **Nilsson et al., 1993**; **von Baumgarten et al., 2008**). In clinical studies, in line with the current study, more CSDs were associated with worse outcome after TBI (**Lauritzen et al., 2011**, **Hartings et al., 2011**; **Chase, 2014**). The suggested mechanism of injury is an ATP-consuming repolarization event following a CSD not followed by a normal hyperemic but by a hypoemic blood flow response. Hypoemia causes a mismatch between blood flow and metabolism thereby inducing or aggravating local tissue ischemia (**Hinzman et al., 2014**). In a previous, as well as in the current, study, we did, however, not find hypoemia but hyperemia after TBI-induced CSD events (**von Baumgarten et al., 2008**). Hence, most likely not tissue ischemia but a direct effect of CSDs on cell swelling or on the permeability of the blood–brain barrier seem to be a more likely scenario how CSDs may cause tissue injury after TBI. That CSDs may directly open the blood–brain barrier has been previously demonstrated (**Gursoy-Ozdemir et al., 2004**) and suggested for a number of neurological disorders (**Takano et al., 2007**; **Dreier and Reiffurth, 2017**; **Helbok et al., 2017**; **Dreier et al., 2018**). Proposed injury mechanisms of CSDs include activation of matrix metalloproteinases (**Gursoy-Ozdemir et al., 2004**), activation of proinflammatory cytokines (**Richter et al., 2017**), and glutamate-mediated effects (**Vinogradova, 2018**; **Crivellaro et al., 2021**; **Menyhárt et al., 2020**), all of which also have been implicated in the pathophysiology of brain edema formation after TBI. Furthermore, glutamate toxicity – a known contributor to posttraumatic brain edema formation – maybe enhanced in FHM1 mice. Hence, the high number of CSDs currently observed in FHM1 mutant mice may be a valid explanation for increased brain edema formation and worse outcome. This increased intracranial hypertension, in turn, reduces cerebral perfusion pressure and thereby indirectly promotes cerebral ischemia. Alternatively, tissue ischemia may occur only in the traumatic penumbra, a narrow volume of tissue surrounding the traumatic contusion, which cannot be assessed with the methodology used. Hence, further studies using techniques with high spatial and temporal resolution, for example, multiphoton microscopy or fluorescence mesoscale imaging, will be needed to further explore vascular and CBF responses in peri- and paralesional brain tissue following TBI.

Interestingly, FHM1 mutant mice showed an increased rate of CSDs in the contralateral, nontraumatized hemisphere. Bihemispheric CSDs have been described in the currently used experimental TBI model, but only immediately after injury and not at later time points (**von Baumgarten et al., 2008**). Since CSDs may theoretically cross to the opposite side of the brain only through brain structures connecting both hemispheres, our results suggest that in FHM1 mutant mice post-trauma CSDs may reach deeper brain structures like the corpus callosum or the striatum. Such a scenario is supported by previous experiments showing that CSDs readily propagate into subcortical structures in FHM1 mutants, specifically in S218L mutants, with an allele-dosage effect (**Eikermann-Haerter et al., 2011**) and by a study showing by diffusion weighted MRI that CSDs migrate into subcortical striatal and hippocampal regions in the same strain of mice (**Cain et al., 2017**). Subcortical propagation of spreading depolarizations into deeper brain regions, even into thalamus in the case of homozygous S218L mutant mice, was proposed as an explanation for how CSDs may increase post-trauma

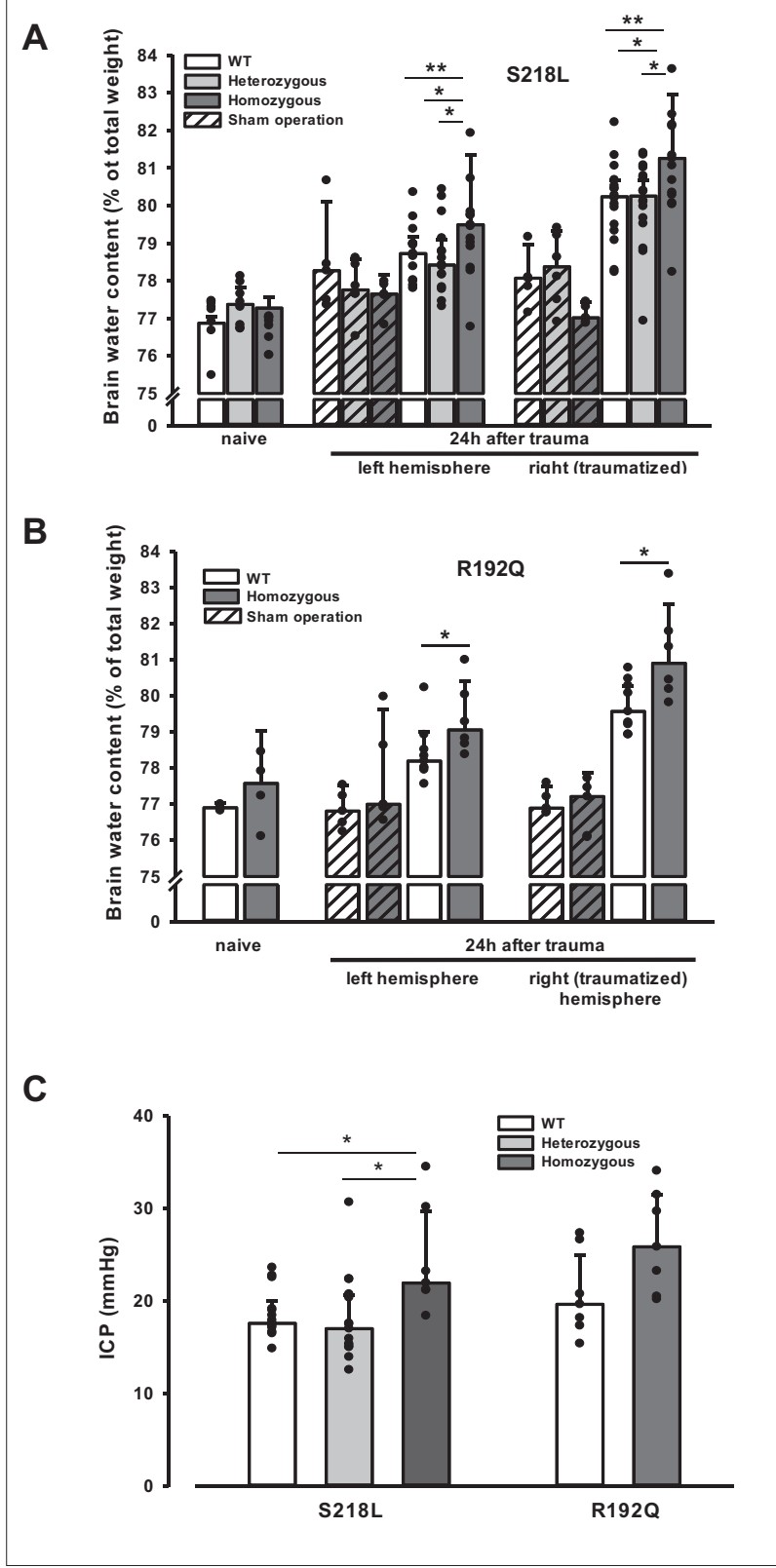

**Figure 4.** Brain edema formation in familial hemiplegic migraine type 1 (FHM1) mutant mice 24 hr after controlled cortical impact (CCI). (**A**) Pretrauma brain water content (*naive*) is approximately 77.5% and comparable in all genotypes; sham operation (striped bars) does not induce significant changes of brain water content. Twenty-four hours after CCI, brain water content in the traumatized (right) hemisphere is significantly increased compared to

*Figure 4 continued on next page*

*Figure 4 continued*

the respective controls; in homozygous S218L mutant mice (dark gray bars) brain edema formation is significantly more pronounced than in wild-type (WT; white bars) and S218L heterozygous mutant mice (light gray bars). Heterozygous animals, in contrast, showed no difference to WT mice. In the contralateral/nontraumatized hemisphere only homozygous S218L mutant mice had a significant increase in brain water content. As in the traumatized hemisphere, brain water content was significantly higher in homozygous S218L mutants than in WT and heterozygous S218L mutant mice (sham/naive: $n$ = 5–8/group; S218L hom./het.: $n$ = 15–18 each; WT: $n$ = 16; median ± 95% confidence interval, *p < 0.05, **p < 0.005 vs. naive brain water content, analysis of variance [ANOVA] on ranks). (**B**) In the R192Q strain, brain water content was within normal limits in naive and sham-operated animals; 24 hr after CCI, however, it was significantly increased in homozygous mutant mice (dark gray bars) in both hemispheres compared to WT (white bars) (Sham/naive: $n$ = 4–5/group; R192Q hom.: $n$ = 6; WT: $n$ = 9; median ± 95% confidence interval, *p < 0.05, rank sum test). (**C**) Intracranial pressure (ICP) 24 hr after trauma was significantly elevated in S218L homozygous mice (left panel, dark gray bar) compared to WT (white bar) and heterozygous mutant mice (light gray bar); in the R192Q strain, ICP tended to be higher in mutant animals (right panel, dark gray bar) than in their WT littermates (white bar; S218L het.: $n$ = 15; S218L hom.: $n$ = 7; S218L WT: $n$ = 15; R192Q hom.: $n$ = 7; R192Q WT: $n$ = 7; median ± 95% confidence interval, *p < 0.005, ANOVA on ranks).

The online version of this article includes the following source data for figure 4:

**Source data 1.** *Figure 4* Brain edema formation after TBI.

mortality in FHM1 mutants. In fact, it was shown recently that in homozygous S218L mutant mice, seizure-related spreading depolarization in the brainstem correlated with respiratory arrest that was followed by cardiac arrest and death (*Loonen et al., 2019*). More precise, spreading depolarizations during spontaneous fatal seizures invaded the respiratory centers of the brainstem to cause apnea (*Jansen et al., 2019*). Whether similar phenomena can explain the observed subacute mortality of homozygous S218L mutant mice after TBI remains unknown and may be difficult to prove experimentally. In conclusion, we demonstrate that FHM1 mutant mice have more severe brain edema and worse outcome after TBI as compared with their WT littermates and that this phenotype is associated with a significant and gene-dose-dependent increase in the number of postinjury CSDs. Hence, we suggest that an increased susceptibility to CSDs is a likely mechanism underlying the unfavorable outcome of FHM1 mutant mice and FHM1 patients following head injury. Furthermore, our results suggest that post-trauma CSDs may induce brain edema formation more directly than previously envisaged, that is without affecting CBF. These mechanistic insights may open the door for further investigations into the mechanism of brain edema formation after TBI.

## Materials and methods

Experiments, group size calculations, and the statistical approach were reviewed by the Animal Review Board and approved by the Veterinary Office of the Government of Upper Bavaria. Group size was calculated with Sigma Stat 3.0 (Jandel Scientific, Erkrath, Germany) using analysis of variance ANOVA and was based on the following parameters: minimal detectable difference between groups: 30%, SD: 15%, power: 0.8, p < 0.05. All acquired data are presented, that is no outliers were removed, and all results are reported in accordance with the ARRIVE 2.0 guidelines.

### Experimental design

Male and female mice with a body weight between 21 and 26 g carrying the S218L (homozygous, heterozygous, WT littermates) or R192Q (homozygous and WT littermates) mutation in the *Cacna1a* gene were used for the current experiments. Animals were enrolled in the study until all groups reached the calculated group size 24 hr after trauma. All mice were genotyped using previously published protocols (*van den Maagdenberg et al., 2004*; *van den Maagdenberg et al., 2010*). An acute group for recording of CSDs for 90 min after TBI and three groups for the assessment of seizure activity, neurological outcome, brain edema, and ICP up to 24 hr after TBI were investigated (*Figure 1A*). Experiments were performed in a randomized manner by drawing lots and researchers were blinded toward the genotype of the animals until the end of data analysis.

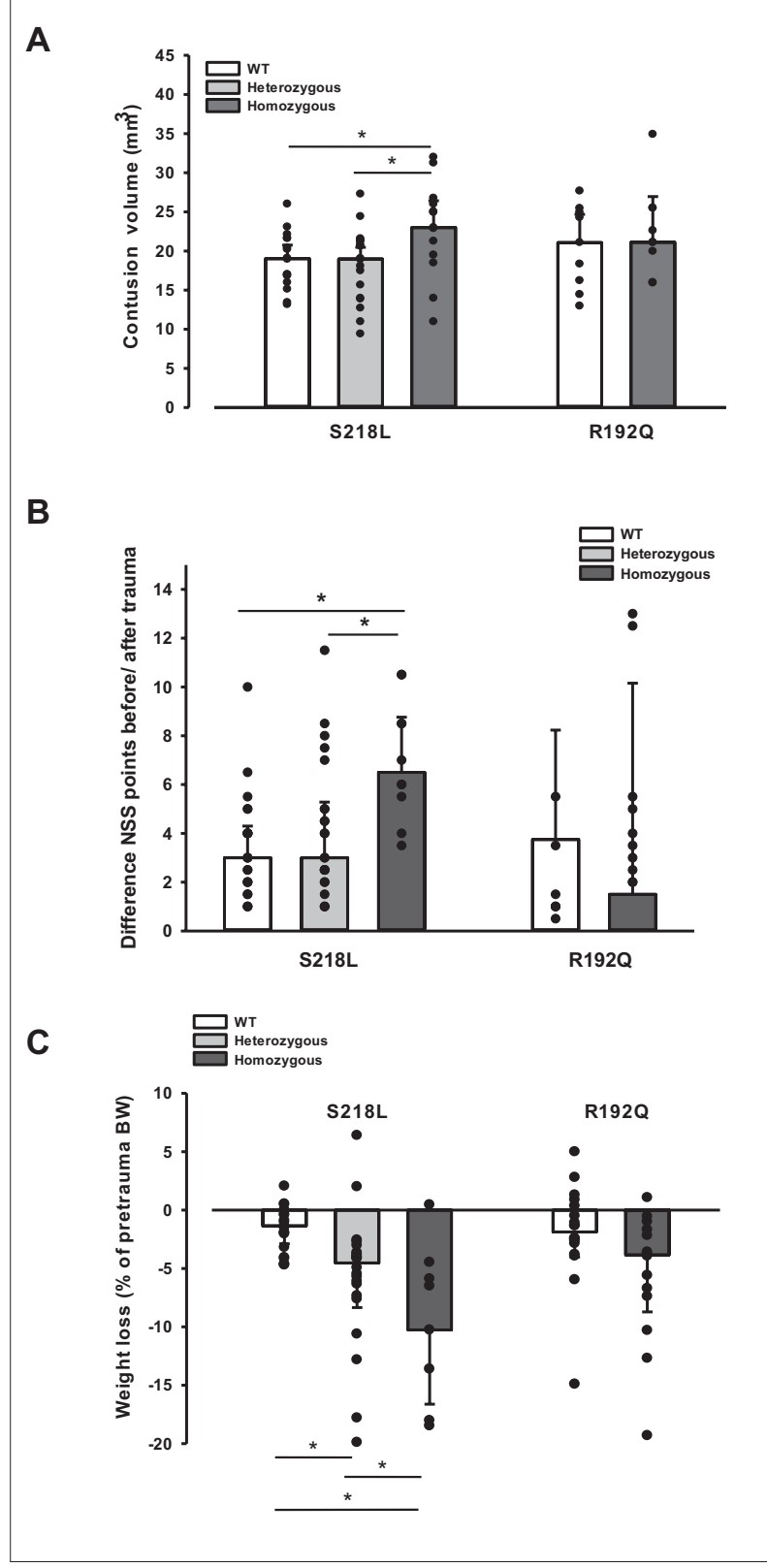

**Figure 5.** Brain damage and functional outcome in familial hemiplegic migraine type 1 (FHM1) mutant mice 24 hr after controlled cortical impact (CCI). (**A**) Volume of structural posttraumatic brain damage, that is contusion volume, was significantly higher in homozygous S218L mutant mice compared to heterozygous S218L mutant and wild-type (WT) animals; in the R192Q strain, lesion volume was only increased by trend (S218L hom./het.: *n* = 13–21

*Figure 5 continued on next page*

*Figure 5 continued*

each; WT: *n* = 16; R192Q hom.: *n* = 9, WT: *n* = 10; median ± 95% confidence interval, *p < 0.5, analysis of variance [ANOVA] on ranks). (**B**) Performance in a multivariate neurological test (maximum deficit 20 points), expressed by differences of points obtained before and 24 hr after traumatic brain injury (TBI). Higher scores indicate higher deficits, that is healthy animals usually score 0–1.5 points. Homozygous S218L mutant mouse's scores deteriorated significantly more than in the heterozygous S218L mutant or WT group (left panel). In the R192Q mutants, performance was comparable to WT mice (S218L hom./het.: *n* = 10–23 each; WT: *n* = 23; R192Q hom.: *n* = 9; WT: *n* = 10; median ± 95% confidence interval, *p < 0.005, ANOVA on ranks, Dunn's post hoc test). (**C**) Weight loss 24 hr after CCI: heterozygous and homozygous S218L mutant mice lost significantly and gene dose dependently more weight, R192Q animals' weight did not differ from their WT littermates (S218L hom./het.: *n* = 9–22; WT: *n* = 21; R192Q hom.: *n* = 14; WT: *n* = 18; median ± 95% confidence interval, *p < 0.05, ANOVA on ranks, Dunn's post hoc test).

The online version of this article includes the following source data for figure 5:

**Source data 1.** *Figure 5* Brain damage and functional outcome in FHM1 mutant mice after TBI.

## Anesthesia and surgical preparations

Mice were anesthetized using an isoflurane chamber (4% isoflurane, 30 s). For trauma induction, anesthesia was continued with 1.2% isoflurane (30% $O_2$/70% $N_2O$) administered via a face mask. In all experiments with neurophysiological recordings anesthesia and monitoring was performed as described previously (*Thal and Plesnila, 2007*). Local CBF was measured above both hemispheres using a laser Doppler probe (Periflux 4,001 Master; Perimed, Stockholm, Sweden) glued onto the skull above the left and the right MCA territory as previously described (*Figure 1B*; *Terpolilli et al., 2009*).

## Trauma induction

TBI was induced using the controlled cortical impact (CCI) model as previously described (*von Baumgarten et al., 2008*; *Terpolilli et al., 2009*; *Terpolilli et al., 2013*). In short, after craniotomy over the right parietal cortex (*Figure 1B*), the lesion was induced with a velocity of 6 m/s, an impact depth of 0.5 mm, and a contact time of 150 ms, settings that result in a cortical contusion (*Schwarzmaier et al., 2015a*, *Schwarzmaier et al., 2015b*). After TBI the bone flap was replaced and fixed with tissue glue.

## Neurophysiological recordings

EEG and DC potential measurements were performed noninvasively using two miniature Calomel electrodes placed onto the skull bone anterior to the contusion and over the contralateral hemisphere and a ground electrode fixed to the parietal scalp (*Figure 1B*) as previously described (*von Baumgarten et al., 2008*).

## Induction of CSD

CSD events were induced by application of 2 µl 0.1 M KCl on the intact dura mater through a burr hole anterior to the contusion (*Figure 1B*) as previously described (*von Baumgarten et al., 2008*). To confirm the preserved ability of the brain to generate CSDs, a KCl-induced CSD was elicited at the end of each experiment.

## Measurement of ICP

ICP was measured in a separate group of mice (*Figure 1A*) 24 hr after trauma by an intraparenchymal probe (Mammendorfer Institut, Mammendorf, Germany) anterior to the contusion as previously described (*Zweckberger et al., 2003*; *Terpolilli et al., 2013*). Briefly, a small burr hole was drilled over the right parietal bone and the ICP probe was lowered into the brain parenchyma. After stabilization of the recordings, ICP was measured over a period of 3 min and averaged.

## Histology and measurement of contusion volume

Contusion volume was determined by histomorphometry using 14 sequential Nissl-stained coronal sections acquired every 500 µm as previously described (*Zweckberger et al., 2003*; *Terpolilli et al., 2013*).

## Measurement of brain water content

Brain water content, a measure for brain edema, was determined 24 hr after trauma using the wet–dry method and is presented as percentage of the total weight of the investigated tissue (*Zweckberger et al., 2006*).

## Neurological outcome

Neurological outcome was assessed 1 hr before and 24 hr after trauma using a modified Neurological Severity Score (NSS; *von Baumgarten et al., 2008*). Zero points indicate no deficit, while 20 points indicate severe neurological deficits.

## Statistical analysis and data presentation

All data were tested for normal distribution with the Kolmogorov–Smirnov test. Based on this analysis, the Mann–Whitney rank sum test was used for comparisons between two groups and ANOVA on ranks (Kruskal–Wallis) followed by Dunn's post hoc test was used for comparing more than two groups. All calculations were performed with a standard statistical software package (Sigma Stat 3.0; Jandel Scientific, Erkrath, Germany). Differences between groups were considered statistically significant at $p < 0.05$. All data are presented as medians ± 95% confidence intervals and individual values.

## Acknowledgements

Data contained in this paper are part of the doctoral thesis of Reinhard Dolp. This study was funded by Munich University FöFoLe Grant #669 (NT), the Netherlands Organization for Scientific Research (NWO) VICI 918.56.602 (MF), the Centre of Medical System Biology (CMSB) in the framework of the Netherlands Genomics Initiative (NGI) 050-060-409 (AvdM), and the European Community (EC) FP7-EUROHEADPAIN (no. 602633; AvdM & MF), the Assistant Secretary of Defence for Health Affairs endorsed by the Department of Defence, through FY 2018 Peer Reviewed Medical Research Program Discovery Award (No. W81XWH1910098; AvdM; Opinions, interpretations, conclusions and recommendations are those of the author and are not necessarily endorsed by the Department of Defence), and by the Deutsche Forschungsgemeinschaft (DFG, German Research Foundation) under Germany's Excellence Strategy within the framework of the Munich Cluster for Systems Neurology (EXC 2145 SyNergy – ID 390857198; NP).

## Additional information

### Funding

| Funder | Grant reference number | Author |
| --- | --- | --- |
| University of Munich FoeFoLe Program | #669 | Nicole A Terpolilli |
| Nederlandse Organisatie voor Wetenschappelijk Onderzoek | 918.56.602 | Arn MJM van den Maagdenberg |
| Centre of Medical System Biology | 050-060-409 | Arn MJM van den Maagdenberg |
| European Commission | FP7-EUROHEADPAIN | Arn MJM van den Maagdenberg |
| Deutsche Forschungsgemeinschaft | EXC 2145 SyNergy - ID 390857198 | Nikolaus Plesnila |
| Defence for Health Affairs, Australia | W81XWH1910098 | Arn MJM van den Maagdenberg |

The funders had no role in study design, data collection, and interpretation, or the decision to submit the work for publication.

## Author contributions
Nicole A Terpolilli, Formal analysis, Investigation, Methodology, Writing – original draft, Writing – review and editing; Reinhard Dolp, Kai Waehner, Elisabeth Rumbler, Investigation; Susanne M Schwarzmaier, Boyan Todorov, Investigation, Methodology; Michel D Ferrari, Conceptualization, Methodology, Resources, Supervision; Arn MJM van den Maagdenberg, Conceptualization, Methodology, Writing – original draft, Writing – review and editing; Nikolaus Plesnila, Conceptualization, Data curation, Formal analysis, Funding acquisition, Project administration, Supervision, Writing – original draft, Writing – review and editing

## Author ORCIDs
Nicole A Terpolilli  http://orcid.org/0000-0001-7070-3113
Nikolaus Plesnila  http://orcid.org/0000-0001-8832-228X

## Ethics
This study was reviewed by the Ethics Board and approved by the Veterinary Office of the Government of Upper Bavaria (protocol # 118/05). All of the animals were handled according to approved institutional animal care protocols. All surgery was performed in deep inhalation anesthesia and animals received appropriate analgesia postsurgery. Every effort was made to minimize suffering.

## Decision letter and Author response
Decision letter https://doi.org/10.7554/eLife.74923.sa1
Author response https://doi.org/10.7554/eLife.74923.sa2

# Additional files

## Supplementary files
• Transparent reporting form

## Data availability
All data generated or analyzed during this study are included in the manuscript. Source data are available for all figures. Raw data are available at the Center for Open Science (OSF): https://osf.io/jqae2.

The following dataset was generated:

| Author(s) | Year | Dataset title | Dataset URL | Database and Identifier |
| --- | --- | --- | --- | --- |
| Plesnila N | 2022 | Mechanisms of brain injury after TBI in FHM1 mice | https://osf.io/jqae2 | Open Science Foundation, jqae |

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
