## [Editor Report]

This paper will be of considerable interest to familial hemiplegic migraine type 1 (FHM) sufferers who may experience traumatic brain injury (and their physicians), as well researchers with an interest in the spectrum and phenotypic consequences of mutations in the voltage-gated, P/Q-type Ca^2+^ channel, *CACNA1A*. The authors demonstrate that patients carrying a gain-of-function S218L missense mutation in *CACNA1A* exhibit a gene-dosage-dependent increase in the susceptibility to cortical spreading depolarization (CSD), seizure activity, and brain edema formation following TBI.

---

## [Decision Letter]

**Decision letter after peer review:**

Thank you for submitting your article "Mutated neuronal voltage-gated Ca_V_2.1 channels causing familial hemiplegic migraine 1 increase the susceptibility for cortical spreading depolarization and seizures and worsen outcome after experimental traumatic brain injury" for consideration by *eLife*. Your article has been reviewed by 2 peer reviewers, one of whom is a member of our Board of Reviewing Editors, and the evaluation has been overseen by Richard Aldrich as the Senior Editor. The reviewers have opted to remain anonymous.

Essential revisions:

1. Following CSDs, the authors describe an interval of hyperemia. However, looking at the CSF graph (Figure 2 A), apparently, there are delayed intervals of reduced blood flow in periods with lower CSD frequency. How do the authors interpret those?

2. Higher mortality in the S218L group is an important finding, however, except that the authors mentioned mice died during generalized tonic-clonic seizures, no further in depth studies have been provided. What would happen if the lesion would be titrated until mortality in S218L mice is zero.

3. Contralateral/belateral CSDs might also be related to variability of lesions, please discuss.

4. As stated in the first sentence of the discussion, the authors aimed at identifying possible mechanisms responsible for worst outcome after brain injury. Despite using transgenic animal models, the authors did not elaborate on cellular resp. molecular mechanisms. Those should at least be discussed.

5. Brain edema does not differ between both transgenic strains (homozygous S218L and R192Q) which also does not explain differences in the outcome measures, please discuss.

6. What was the rationale to measure ICP only at the endpoint of study and not over a longer period of time during the first 24 hours after CCI onset?

---

## [Author Response]

Essential revisions:1. Following CSDs, the authors describe an interval of hyperemia. However, looking at the CSF graph (Figure 2 A), apparently, there are delayed intervals of reduced blood flow in periods with lower CSD frequency. How do the authors interpret those?

Thank you for this comment. TBI induces brain swelling and subsequent intracranial hypertension. Intracranial hypertension reduces cerebral perfusion pressure and, hence, cerebral blood flow (in the whole brain) as we previously demonstrated [1, 2]. This is exactly what was also observed in the experiments shown in figure 2A: between CSDs CBF decreased in both hemispheres below baseline values. On top of this ICP-induced global reduction of CBF, CSDs induced hyperemia.

2. Higher mortality in the S218L group is an important finding, however, except that the authors mentioned mice died during generalized tonic-clonic seizures, no further in depth studies have been provided.

The finding that mice died during or shortly after a tonic-clonic seizure was obtained by clinical diagnosis, i.e. mice were observed by several investigators during the first 24 hours after TBI. A tonic-clonic seizure in a mouse is quite impressive and can therefore easily by diagnosed without any further analysis. Animals were unconscious during the post-ictal period and died during this period of unconsciousness by respiratory arrest. Based on this very clear clinical phenotype we decided not to perform any on depth analysis regarding the cause of death because such an analysis was outside the scope of study and would have required more transgenic mice, which were extremely hard to breed. We fully agree that the cause of death would be of interest, but in view of the high number of animals needed to perform such an investigation properly and the difficulties to breed the S218L line, we decided to use the available transgenic mice for the other experiments necessary to address the aims of the current study.

What would happen if the lesion would be titrated until mortality in S218L mice is zero.

This is a very interesting, but quite hypothetical question, since a quite large number of mice would be required to provide experimental proof for any statement on this issue. As stated above, the main cause of mortality are tonic-clonic seizures. Such seizures are caused, like CSDs, by a reduced threshold for neuronal network activity. Since S218L mice have such a reduced threshold for neuronal network activity that even minimal manipulations at the skull/meninges elicit CSDs (see our data on CSD before TBI), it is quite probable that any kind of mild or moderate trauma to the brain will cause tonic-clonic seizures. Therefore, my best guess is that any trauma intensity will result in tonic-clonic seizures and mortality in these animals. This is exactly what is also observed in patients carrying the S218L mutation: they develop massive brain swelling and may eventually die even after very minor head trauma. Hence, our mouse model replicates also this aspect of the human pathology.

3. Contralateral/belateral CSDs might also be related to variability of lesions, please discuss.

We can exclude with a relative high level confidence that an *a priori* lesion variability is the cause for contralateral/bilateral CSDs. The reason for this confidence is that the Controlled Cortical Impact TBI model used in the current study is extremely reproducible in terms of lesion size and location. The piston, which injures the cortex, is located on the surface of the brain with the help of a stereotactic device with an accuracy of 0.1 mm. The lesion itself shows a variability of only 5% (SD in % of the mean lesion volume of 8-12 mice). Usually biological models have variabilities of 20% and more. In our opinion there are therefore no reasons to believe that the variability of the TBI model itself is the reason for the observed contralateral/bilateral CSDs.

The best explanation we can offer why not all mice show contralateral/bilateral CSDs is the genotype of the S218L transgenic animals. We think that the increased edema formation in these animals leads to narrowing of the interhemispheric fissure facilitating the spread of a CSD to the contralateral hemisphere; also, the lower threshold and increased propagation velocity of CSDs in transgenic animals may lead to development of subcortical CSDs that travel along subcortical structures like the corpus callosum to the contralateral side. These CSD events then may promote lesion progression, resulting in a higher lesion size variability.

4. As stated in the first sentence of the discussion, the authors aimed at identifying possible mechanisms responsible for worst outcome after brain injury. Despite using transgenic animal models, the authors did not elaborate on cellular resp. molecular mechanisms. Those should at least be discussed.

Thank you very much. We edited the manuscript as following.

“Proposed injury mechanisms of CSDs include activation of matrix metallo-proteinases [3], activation of pro-inflammatory cytokines [4], and glutamate-mediated effects [5-7], all of which also have been implicated in the pathophysiology of brain edema formation after TBI. Furthermore, glutamate toxicity – a known contributor to posttraumatic brain edema formation – is most probably enhanced in FHM1 mice, which display a reduced threshold for neuronal activity. This is very much supported by the observed mortality during tonic-clonic seizures”

5. Brain edema does not differ between both transgenic strains (homozygous S218L and R192Q) which also does not explain differences in the outcome measures, please discuss.

Brain edema formation was assessed 24h after TBI, i.e. only in animals which survived until this time point. Animals which died before brain edema was measured, died most likely from seizures and excessive brain edema formation. Thus, the brain water content obtained 24h after TBI suffers from a high mortality bias. This interpretation is supported by brain water content measurement performed in single animals who died while being observed: values were much higher than in mice surviving until 24 hours.

6. What was the rationale to measure ICP only at the endpoint of study and not over a longer period of time during the first 24 hours after CCI onset?

Thank you for this comment. ICP measurements are performed with an intraparenchymal sensor and therefore requires anesthesia. In patients telemetric ICP sensors are under development, however, such wireless devices are rather large and not suitable for mice.

References

1. Terpolilli, N.A., et al., Inhaled nitric oxide reduces secondary brain damage after traumatic brain injury in mice. J. Cereb. Blood Flow Metab, 2013. 33(2): p. 311-318.

2. Terpolilli, N.A., et al., The novel nitric oxide synthase inhibitor 4-amino-tetrahydro-L-biopterine prevents brain edema formation and intracranial hypertension following traumatic brain injury in mice. J Neurotrauma, 2009. 26(11): p. 1963-75.

3. Gursoy-Ozdemir, Y., et al., Cortical spreading depression activates and upregulates MMP-9. J Clin Invest, 2004. 113(10): p. 1447-55.

4. Richter, F., et al., Effects of interleukin-1ß on cortical spreading depolarization and cerebral vasculature. J Cereb Blood Flow Metab, 2017. 37(5): p. 1791-1802.

5. Menyhárt, Á., et al., Malignant astrocyte swelling and impaired glutamate clearance drive the expansion of injurious spreading depolarization foci. J Cereb Blood Flow Metab, 2021: p. 271678x211040056.

6. Crivellaro, G., et al., Specific activation of GluN1-N2B NMDA receptors underlies facilitation of cortical spreading depression in a genetic mouse model of migraine with reduced astrocytic glutamate clearance. Neurobiol Dis, 2021. 156: p. 105419.

7. Vinogradova, L.V., Initiation of spreading depression by synaptic and network hyperactivity: Insights into trigger mechanisms of migraine aura. Cephalalgia, 2018. 38(6): p. 1177-1187.